# Fast Conflict Detection for Multi-Dimensional Packet Filters

**Chun-Liang Lee [1], Guan-Yu Lin [2,*] and Yaw-Chung Chen [2]**

1   Department of Computer Science and Information Engineering, School of Electrical and Computer Engineering, College of Engineering, Chang Gung University, Taoyuan 33302, Taiwan
2   Department of Computer Science, National Yang Ming Chiao Tung University, Hsinchu 30010, Taiwan
*   Correspondence: guanyu@cs.nctu.edu.tw

**Abstract:** To support advanced network services, Internet routers must perform packet classification based on a set of rules called packet filters. If two or more filters overlap, a filter conflict will occur and lead to ambiguity in packet classification. Further, it may affect network security or even the correctness of packet routing. Hence, it is necessary to detect conflicts to avoid the above problems. In recent years, many conflict detection algorithms have been proposed, but most of them detect conflicts for only prefix fields (i.e., source/destination IP address fields) of filters. For greater practicality, conflict detection must include non-prefix fields such as source/destination IP port fields and the protocol field. In this study, we propose an efficient conflict detection algorithm for five-dimensional filters, which include both prefix and non-prefix fields. In the proposed algorithm, a tiny lookup table is created for quickly filtering out a large portion of non-conflicting filter pairs, thereby reducing the overall conflict detection time. Experimental results show that our algorithm reduces the detection time by 10% to 28% compared with other conflict detection algorithms for 20 K filter databases. More importantly, our algorithm can be used to extend any existing conflict detection algorithms for two-dimensional filters to support fast conflict detection for five-dimensional filters.

**Keywords:** conflict detection; firewall policy; packet classification; packet filters; network security

## 1. Introduction

With the rapid growth of the Internet, many advanced network services such as firewalls, differentiated services, policy-based forwarding, quality of services (QoS), and network security have been developed. To support these services, packet classification plays a crucial role in the Internet [1–4]. Packet filters are the rules that routers use to classify incoming packets based on the header information. For supporting various network services, a filter typically contains five fields, including the source/destination IP addresses, the source/destination ports, and the protocol type [5]. An IP address field usually indicates a prefix. An IP address that matches a prefix indicates that the IP address contains the same prefix content. A port field usually indicates a range of values. For example, a port value $v$ matches a range of [$s$:$e$] if $s \leq v \leq e$. The protocol field usually indicates an exact value. When the field content is expressed using *, it is regarded as a wildcard rule, indicating that the value of the field covers the entire range. For a packet $P$ and a filter $F$, we can say that $P$ matches $F$ if the corresponding fields of $P$ match all fields of $F$ [1].

Packet classification ambiguity occurs when a packet $P$ matches two or more filters but the associated actions of matching filters are different [6]. In this situation, the classifier cannot correctly determine the actions that should be taken on $P$. This may lead to incorrect packet classification and cause security vulnerabilities of firewall-similar services as well as data routing errors [6–9]. When the aforementioned problems occur, a heavy burden is thrust upon Internet users. These problems also affect the transmission efficiency and reliability of the entire network. Therefore, it is essential to detect conflicts in a filter database to prevent incorrect classification [10]. An earlier study listed three possible solutions [6] that are accomplished by means of the following:

1. Select the first filter in the database that matches $P$.
2. Assign each filter a priority. From the set of filters that match $P$, select the filter having the highest priority.
3. Assign each field a priority. From the set of filters that match $P$, select the filter having the most specific field with the highest priority.

However, these solutions cannot fully resolve conflicts. We use the two-dimensional (2D) filters in Table 1 for illustration. Suppose that the length of each field is 4 bits. The rectangles in Figure 1 are drawn according to the filters in Table 1, with overlapping regions indicating the conflict regions between filters. We give priority to the two conflicting filters and define $F \rightarrow G$ to indicate that the priority of filter $F$ is higher than that of filter $G$. If a packet matches both filters $F$ and $G$, the packet will execute the associated action of $F$. The priorities of the four overlapping filters in Figure 1 are set as follows: $a \rightarrow b$, $b \rightarrow c$, $c \rightarrow d$, and $d \rightarrow a$. However, the order of these four filters causes a cycle relationship. In this case, we cannot prevent conflicts by setting priorities because we cannot sort these four filters in an absolute order. Therefore, Hari et al. [6] proposed a method of adding resolve filters to solve filter conflicts. In other words, the method adds a new filter to the overlapping region of two filters and assigns it a high priority to break the cycle. For example, adding a resolve filter $e$: (001*, 001*) to the overlapping region of $a$ and $b$, and setting the priority of $e$ higher than that of $a$ and $b$, solves the problem of filter conflicts.

**Table 1.** An example of a 2D filter database.

| Filter | Source IP Address | Destination IP Address | Action |
|---|---|---|---|
| $a$ | 001* | * | Accept |
| $b$ | * | 001* | Reject |
| $c$ | 110* | * | Accept |
| $d$ | * | 110* | Reject |

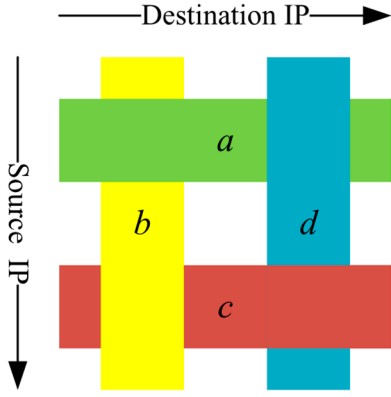

**Figure 1.** Rectangular presentation of filters.

Although filter conflicts can be resolved by adding resolve filters, it is necessary to update the database which affects the effectiveness of database updates. Some Internet service applications such as intrusion detection, stateful filtering, and customer relationship management may frequently add or update filters [7,11], which may conflict with existing filters in the filter database. Therefore, a conflict detection algorithm must be designed to determine all potential conflicts. This is necessary to ensure the security and user QoS whenever the database is updated with any filters, even when the new filters may not produce conflicts [12].

As Internet Protocol version 6 (IPv6) gradually gains popularity and the number of filters becomes greater, it is expected that new advanced network services will constantly emerge and dynamic filter updates will be more frequent. In addition, in the new generation network, architecture known as a software-defined network (SDN) [13], 12 or more filter fields must be compared. Packet classification technology must be able

to cope with these great changes [14]. To improve the efficiency of packet classification, many packet classification algorithms have been proposed and they can be divided into two implementation types: software-based [3,11,14–18] and hardware-based [4,19–23]. Because conflict detection can ensure the correctness of packet classification, the conflict detection process must be performed whenever filters are updated. In this case, conflict detection performance must also be considered; otherwise, it will become the bottleneck of router performance.

The majority of existing conflict detection algorithms can only detect conflicts for 2D filters. For greater practicality, the following three non-prefix fields must be considered: source port, destination port, and protocol fields. To extend conflict detection from two to more than two dimensions, the first problem encountered is the inconsistent data formats of fields. Using the same data structures to store the information of different fields is difficult. When processing these non-prefix fields, the previous works restrict the field to be either exact values or a wildcard [6], or by converting all the fields into prefixes [7]. However, the port field is usually represented as a range, and range-to-prefix conversion may cause filter replication. For example, suppose that the longest prefix length is 4, the range [1:14] can be represented by as many as six prefixes: 0001, 001*, 01*, 10*, 110*, and 1110. If the port field value of a filter is [1:14], this filter must have six replications so that six new filters can indicate the same range as the original filter. As the number of filters increases, the filter replication may cause the number of filters to grow exponentially. As a result, not only larger memory space but also longer time are required to perform the conflict detection.

The objectives of this study are as follows: (1) to analyze the characteristics of non-prefix fields in five-dimensional (5D) filters; (2) to investigate the rules for determining whether two 5D filters conflict based on the results of the prefix field comparisons and the non-prefix field comparisons; and (3) to propose an efficient algorithm that can be used either to extend an existing 2D conflict detection algorithm to support 5D filters, or to increase the throughput of an existing 5D conflict detection algorithm.

In our proposed algorithm, we divide the 5D conflict detection process into two parts. First, we obtain comparison combinations of two filters in prefix fields then, based on these combinations, we sum up the matching combinations of non-prefix fields to ensure that conflicts occur after combined detection. We classify the common values of non-prefix fields into several sets, then discuss the relationship between different sets and analyze the comparison results of every combination. Finally, we create a lookup table based on each comparison result. When we detect the non-prefix fields of filters, the table checking process can filter out the combinations with known results to reduce the number of detected filter pairs and thus reduce the required time of conflict detection.

The remainder of this paper is organized as follows. Section 2 reviews the existing conflict detection algorithms. Section 3 defines the conditions for 5D filter conflicts. Section 4 presents our key research ideas and explains how to construct the data structure and execution mode of our algorithm. Section 5 describes our experimental procedures and results. Section 6 concludes this study.

## 2. Related Work

To detect all filter conflicts, a straightforward approach is to compare every pair of filters in the filter database. Obviously, this approach is simple and does not require extra storage. However, it takes $O(n^2)$ time to detect all conflicts, where $n$ is the number of filters, which is not feasible for large filter databases. Thus, several conflict detection algorithms have been proposed in recent years. In this section, we divide these algorithms into two groups based on whether they can be directly extended to 5D conflict detection.

### 2.1. Conflict Detection for 5D Filters

Hari et al. [6] first defined the source IP address and destination IP address fields as 2D prefix fields and introduced the notion of 2D filter conflict. Two filters, *F* and *G*, are in conflict if and only if one of the following conditions holds:

1. *F*[1] is a prefix of *G*[1] and *G*[2] is a prefix of *F*[2], or
2. *G*[1] is a prefix of *F*[1] and *F*[2] is a prefix of *G*[2].

*F*[1] and *G*[1] indicate the first prefix field of *F* and *G*, respectively, whereas *F*[2] and *G*[2] indicate the second prefix field of *F* and *G*, respectively.

Hari et al. proposed the *FastDetect* algorithm [6] using the Grid-of-Trie and a switch pointer [24] for 2D conflict detection. All conflicting filter pairs can be reported in $O(nW + S)$ time, where $W$ is the length of the longest prefix and $S$ is the number of conflicting filter pairs. The memory requirement of the *FastDetect* algorithm is $O(nW)$. When the *FastDetect* algorithm is extended for 5D conflict detection, for non-prefix fields, Hari et al. assumed the port fields of a filter must be either an exact value or a wildcard, while the protocol field of a filter must be one of the following: transmission control protocol (TCP), user datagram protocol (UDP), or a wildcard. In case a filter whose protocol field is a wildcard, the filter will be replicated three times: once with the protocol field set to TCP, once to UDP, and once to the special value OTHER. Finally, the filter sets are divided into sets of TCP, UDP, and OTHER according to the protocol values, thus ensuring that filters in different sets do not conflict [6]. Based on these assumptions, the number of filters that require comparison during conflict detection can be reduced and it does not require extra auxiliary information. However, filter replication caused by the aforementioned method will consume considerable memory space. In addition, the port and protocol field values can only be exact values or a wildcard, and clearly this does not conform to a practical application.

Baboescu and Varghese [7] proposed the scalable bit vector (SBV) conflict detection algorithm based on the bit vector (BV) scheme [25] and the aggregated bit vector scheme [26]. They proposed to use compressed binary tries (each field bit requires the use of a trie) and $n$ bit vectors to determine dynamically whether new filters conflict with the original filter sets in $O(nW)$ time. Their proposed algorithm can detect all conflicts with $O(knW)$ time and $O(kn^2)$ space, where $k$ indicates the number of fields in a filter. In the study [7], all the non-prefix fields are converted into the prefix fields by using the method described in [26] during 5D conflict detection; this method causes many filter replications and requires considerable memory space. Lai and Wang [10] proposed several algorithms that modified the original BV scheme to prevent the defect of massive memory duplication caused by conversion from range to prefix and developed a method for comparing range fields. This allows the algorithm to support 5D conflict detection. However, it remains costly in terms of memory requirements due to the high cost of the auxiliary information used to compare the range fields. Kuo et al. [27] proposed a compact bit vector (CBV) conflict detection algorithm to improve the constructed matching tries of the SBV algorithm. In the CBV algorithm, they proposed a redundancy reduction scheme and exploited the covering and potential conflict relationships between filters to significantly reduce the number of filters that must be involved in the construction of matching tries. The CBV algorithm then further merged the redundant match nodes in each matching trie by adopting an upward merging approach. Finally, the highly compact matching tries were built to represent the relationships between filters. However, the CBV algorithm does not consider the case where the non-prefix field is represented as an ambiguity range. Similar to [7,10], using the trie-based data structure incurs a high memory requirement, and a highly compact scheme will degrade the performance of updating filters.

*2.2. Conflict Detection for 2D Filters*

Lu and Sahni [12] determined that when a 2D filter is represented by a geometric area, two conflicting filters will generate an overlapping area in the plane, and at least one perfect crossing among the segments of two areas must exist. Lu and Sahni defined perfect crossing as two line segments of the share point perfectly crossing if and only if they cross and the crossing point is not an endpoint of either line segment. In addition, they proposed a magnifying mechanism to ensure that each filter conflict has the characteristics of perfect crossing. They then used Bentley's and Ottmann's algorithm [28] to detect all

existing perfect crossings and finally to determine all conflicts. The time complexity of this algorithm is $O(n\log n + S)$ and the space complexity is $O(n)$. Lee et al. [29] proposed a 2D tuple space search algorithm (TCDA) based on a tuple space data structure and hash search. The execution of conflict detection in each filter can be accelerated by adding a marker pointer and filter pointer. The TCDA can determine all conflicts in $O(nW + S)$ time. Kwok and Poon [30] studied the 2D packet conflict problem and discovered that the problem can be reduced to the persistent predecessor problem. They used a balanced binary search tree to perform 2D conflict detection, with the time complexity further improved to $O(n \min\left\{\frac{\log w \log \log n}{\log \log w}, \sqrt{\frac{\log n}{\log \log n}}\right\} + S)$. Maindorfer et al. [31] converted all arbitrary 1D range filters into consecutive slabs and defined all conflict relationships between all consecutive slabs. They used a slab-detect algorithm and proposed an output-sensitive algorithm that was able to detect all filter conflicts in $O(n\log n)$ time with the space complexity being $O(n)$. Zhang et al. [9] used a formal method to analyze the meaning of IPv6 firewall filters and took the formal validation tool (satisfiability modulo theories solver Z3) [32] to find all the conflicts between every two firewall filters.

These 2D conflict detection algorithms cannot be directly extended for 5D conflict detection due to the limitation of the constructed data structures. More specifically, it is difficult or even impossible to represent 5D filters as geometric areas [12], tuple spaces [29], or binary trees [30]. A simple way to make a 2D conflict detection algorithm capable of handling 5D filters is to compare prefix fields and non-prefix fields separately. However, without a good design of data structure and algorithm for non-prefix fields, it is difficult to achieve a satisfactory detection speed.

## 3. Definition of 5D Filter Conflict

In this section, we define the relationship between two filters in this study. A *d*-dimensional filter database indicates that every filter in the database contains *d* fields. $F[k]$ indicates the value of the *k*th field of filter *F*, $1 \leq k \leq d$. $F[k] \subset G[k]$ indicates that $F[k]$ must be a strict subset of $G[k]$. For example, prefix 100* is a strict subset of prefix 10* and range [0:1023] is a strict subset of range [0:65535]. However, range [0:1023] is not a strict subset of range [1000:2000]. $F[k] \subseteq G[k]$ indicates that $F[k]$ is a general subset of $G[k]$. The difference between general and strict subsets is that the relationship of $F[k] = G[k]$ is also contained by the general subset. $F[k] \cap G[k] = \varnothing$ indicates that the intersection of $F[k]$ and $G[k]$ is the empty set. In other words, we cannot find a value that matches both $F[k]$ and $G[k]$. Therefore, *F* does not conflict with *G* in this condition. In this study, we define 5D conflict detection for five fields of a filter, including two prefix fields (source/destination IP address) and three non-prefix fields (source/destination port and the protocol).

If the conflict conditions defined by [6] are extended for 5D conflict detection, a conflict occurs between *F* and *G* if and only if the following two conditions hold:

**Condition 1.** $F \cap G \neq \varnothing$.

This condition comes directly from the definition of filter conflict. More specifically, this condition can be expressed as $\forall_{1 \leq k \leq 5} F[k] \cap G[k] \neq \varnothing$.

**Condition 2.** $\exists_{1 \leq i \leq 5} \exists_{1 \leq j \leq 5}((i \neq j) \wedge (F[i] \subset G[i]) \wedge (F[j] \supset G[j]))$.

This condition states that there exist two fields, say *i*th and *j*th fields, such that $F[i]$ is a strict subset of $G[i]$ and $G[j]$ is a strict subset of $F[j]$. This condition excludes the case where *F* is a subset of *G* or vice versa.

## 4. An Efficient Conflict Detection Algorithm for Non-Prefix Fields

For 5D conflict detection, we notice that when Condition 1 defined in Section 3 is satisfied, in the five fields, up to ten ($C_2^5$) types of two-field $(i, j)$ combinations exist to satisfy Condition 2 defined in Section 3. To determine the two fields to satisfy Condition 2, an extremely complex execution process is required. Moreover, since the fields of filters are stored in different data formats, it is difficult to design an effective data structure

and algorithm to detect all conflicts. Therefore, if the existing algorithms are extended to detect 5D conflicts, the performance of most of them will be seriously affected. In order to simplify conflict detection, the key idea of our study is to divide the conflict detection into the detection of prefix fields and detection of non-prefix fields. We firstly analyze the comparison result combinations of filters $F$ and $G$ in prefix fields, then match them with the combinations of non-prefix fields of $F$ and $G$ to report conflict with our pre-computed information, which can reduce the number of comparisons of non-prefix fields and further speed up the execution of 5D conflict detection.

The remainder of this section is organized as follows: Section 4.1 introduces our combined detection method, which can reduce the complexity of 5D conflict detection by analyzing all combinations for 5D filter conflicts. Section 4.2 explores the characterization and classification of non-prefix fields and defines the results of comparing two filters in a single non-prefix field. Section 4.3 explores the results of comparing all non-prefix fields and describes how to use our pre-computed information to reduce conflict detection for non-prefix fields. Section 4.4 describes how to construct a lookup table based on our pre-computed information and illustrates how to extend the existing 2D algorithm to 5D conflict detection with our proposed algorithm. The database in Table 2 is used as a supplementary explanation.

**Table 2.** An example of a 5D filter database.

| Filter | Source IP Address | Destination IP Address | Source Port | Destination Port | Protocol |
|---|---|---|---|---|---|
| $F_1$ | 000* | 01* | 80 | * | TCP |
| $F_2$ | 100* | 01* | * | * | TCP |
| $F_3$ | 100* | 01* | * | 60 | * |
| $F_4$ | 10* | 0* | * | 0–1023 | TCP |
| $F_5$ | 00* | 011* | 0–1023 | * | * |
| $F_6$ | 00* | 011* | 1024–65,535 | * | UDP |
| $F_7$ | 0* | 0* | * | * | TCP |
| $F_8$ | * | 000* | * | 1000–5000 | * |

*4.1. Combined Detection Method for 5D Conflict Detection*

The comparison results of prefix fields $x$ and $y$ of $F$ and $G$ can be divided into the following three combinations except for the empty set (Condition 1 defined in Section 3):

**Combination 1.** $(F[x] \subset G[x]) \wedge (F[y] \supset G[y])$

Because the comparison results of $F$ and $G$ in the prefix fields already satisfy Condition 2 defined in Section 3, the conflict conditions can be reached after the final combined detection, when the comparison results of non-prefix fields are not empty. The prefix field comparison results of filter pairs $(F_1, F_5)$, $(F_1, F_6)$, $(F_4, F_8)$, and $(F_7, F_8)$ in Table 2 belong to this type. After combined detection, $(F_1, F_5)$, $(F_4, F_8)$, and $(F_7, F_8)$ are conflicting filter pairs. By contrast, filter pair $(F_1, F_6)$ is not in conflict because $F_1[Protocol] \cap F_6[Protocol] = \varnothing$.

**Combination 2.** $(F[x] \subseteq G[x]) \wedge (F[y] \subset G[y])$

In this combination, the values of the prefix field of $F$ are the strict subset of $G$, or the comparison results of $F$ and $G$ are equal in field $x$, whereas $F[y]$ is the strict subset of $G[y]$. To reach conflict conditions after the combined detection, the comparison results of $F$ and $G$ in non-prefix fields must not be empty, and at least a field $s$ causes the relationship $F[s] \supset G[s]$ to be valid. The prefix field comparison results of filter pairs $(F_1, F_7)$, $(F_2, F_4)$, $(F_3, F_4)$, $(F_5, F_7)$, and $(F_6, F_7)$ in Table 2 belong to this type. After combined detection, $(F_2, F_4)$, $(F_3, F_4)$, and $(F_5, F_7)$ are conflicting filter pairs. Although the comparison results of filter pair $(F_1, F_7)$ in non-prefix fields are not empty, no field $s$ causes the relationship $F[s] \supset G[s]$ to be valid. Similarly, filter pair $(F_6, F_7)$ is not in conflict because $F_6[Protocol] \cap F_7[Protocol] = \varnothing$.

**Combination 3.** $(F[x] = G[x]) \wedge (F[x] = G[x])$

In this combination, all values of prefix fields of $F$ are equal to those of $G$. To reach conflict conditions after combined detection, the comparison results of $F$ and $G$ in non-prefix fields must not be empty, and at least two fields $s$ and $t$ cause the relationship $(F[s] \supset G[s]) \wedge (F[t] \subset G[t])$ to be valid. The prefix field comparison results of filter pairs $(F_2, F_3)$ and $(F_5, F_6)$ in Table 2 belong to this type. Because filter pair $(F_2, F_3)$ contain fields Destination port and Protocol, which produce $F_2[Protocol] \subset F_3[Protocol]$ and $F_2[Destination\ port] \supset F_3[Destination\ port]$, filter pair $(F_2, F_3)$ is in conflict. Because $F_5[Source\ port] \cap F_6[Source\ port] = \varnothing$, $(F_5, F_6)$ is not in conflict.

Based on the analytical results, all conflicting filter pairs can be divided into the three combinations listed in Table 3. The sets of non-prefix field combinations of Combinations 1–3 are expressed as $3Dset1$, $3Dset2$, and $3Dset3$, and the relationship between these sets is $3Dset1 \supset 3Dset2 \supset 3Dset3$. From this relationship, we can determine the following:

1.  When the non-prefix field combinations belong to $3Dset3$, it means that we can find any two non-prefix fields $s$ and $t$ of filters $F$ and $G$ which satisfy Condition 2 defined in Section 3. Regardless of the combination to which the prefix field comparison results belong (except for the empty set), conflicts occur after combined detection.
2.  When the non-prefix field combinations belong to $3Dset2$, regardless of whether the prefix field comparison results belong to Combination 1 or 2, conflicts occur after combined detection because we can find at least one prefix field $x$ and non-prefix field $s$ of filters $F$ and $G$ which satisfy Condition 2.
3.  When the non-prefix field combinations belong to $3Dset1$, conflicts may not occur after combined detection with the prefix field comparison results that belong to Combination 2 because some non-prefix field combinations belong to $3Dset1$ but do not belong to $3Dset2$. Similarly, when the non-prefix field combinations belong to $3Dset2$, conflicts may not occur after the combined detection with the prefix field comparison results that belong to Combination 3. In this situation, we may not find any two fields $i$ and $j$ of filters $F$ and $G$ which satisfy Condition 2.

**Table 3.** All combinations for 5D filter conflict.

| Results of Prefix Fields of $F$ and $G$ | Corresponding to the Combination of Non-Prefix Fields |
| --- | --- |
| Combination 1 | $3Dset1 : \forall_{3 \leq r \leq 5}(F[r] \cap G[r]) \neq \varnothing$ |
| Combination 2 | $3Dset2 : \forall_{3 \leq r \leq 5}(F[r] \cap G[r]) \neq \varnothing \wedge \exists_{3 \leq r \leq 5}(F[s] \supset G[s])$ |
| Combination 3 | $3Dset3 : \forall_{3 \leq r \leq 5}(F[r] \cap G[r]) \neq \varnothing \wedge \exists_{3 \leq s \leq 5}\exists_{3 \leq t \leq 5}((s \neq t) \wedge (F[s] \supset G[s]) \wedge (F[t] \subset G[t]))$ |

Our combined detection method analyzed all combinations for 5D filter conflict, which can reduce the complexity of the original 5D conflict detection according to the predefined information of prefix fields and non-prefix fields sets. It can also be applied to any existing 2D filter conflict detection algorithm that extends to 5D conflict detection because the combined detection method compares non-prefix fields based on the results of prefix-field comparison.

### 4.2. Characteristic Analysis and Classification for Non-Prefix Field

Without any pre-computed information, complete information regarding non-prefix fields of the two to-be-compared filters must be fetched prior to non-prefix field conflict detection, and the results of combined detection can be determined after a series of logic comparison processes. We call this an exact comparison. If exact comparison is used during every combined detection, it may cause too many memory accesses and increase the detection time. If the results of combined detection can be obtained based on the pre-computed information prior to executing an exact comparison, then only part of the combined detections require exact comparison, hence the number of memory accesses can be reduced. Therefore, we analyze the characteristics of the non-prefix field and use the

rarely pre-computed information to design a filtering algorithm so that the number of exact comparisons can be reduced.

According to [33], the values of the source/destination port field can be divided into the following five classes, while the values of the protocol field can be divided into the following four classes, as shown in Tables 4 and 5, respectively:

**Table 4.** Five classes of the source/destination port field.

| Value | Description |
|---|---|
| WC | Wildcard (*) |
| HI | Ephemeral user port range [1024:65535] |
| LO | Well-known system port range [0:1023] |
| AR | An arbitrary range, for example [1000:5000] |
| EM | Exact value, for example [21:21] |

**Table 5.** Four classes of the protocol field.

| Value | Description |
|---|---|
| WC | Wildcard (*) |
| TCP | Transmission control protocol |
| UDP | User datagram protocol |
| Other | The other protocol values, for example ICMP, ESP, etc. |

When a port field value is a type of AR or EM, it can be classified as an HI or LO based on its covered region. For example, if a field value belongs to an EM category and has an exact value of 23, it can be classified as an LO; if the field value is in an AR class and has an arbitrary range [2000:5000], it can be classified as an HI. However, a special case exists in which the AR category field value covers both HI and LO regions (e.g., number range [20:5000]). Since it cannot be exactly classified as HI or LO, we classify the field values of this classes as a new class called Both. Finally, the port field values can be classified into four new classes: WC, HI, LO, and Both. According to the statistics in [24], the common values of protocol fields contain TCP, UDP, and wildcard. The other protocol values are rare, so we can classify them as a new class called Other. Finally, the protocol field values are also classified into four classes: WC, TCP, UDP, and Other. Table 6 shows the classification results of the database in Table 2.

**Table 6.** After classification based on each non-prefix field value in Table 2.

| Filter | Source IP Address | Destination IP Address | Source Port | Destination Port | Protocol |
|---|---|---|---|---|---|
| $F_1$ | 000* | 01* | LO | WC | TCP |
| $F_2$ | 100* | 01* | WC | WC | TCP |
| $F_3$ | 100* | 01* | WC | LO | WC |
| $F_4$ | 10* | 0* | WC | LO | TCP |
| $F_5$ | 00* | 011* | LO | WC | WC |
| $F_6$ | 00* | 011* | HI | WC | UDP |
| $F_7$ | 0* | 0* | WC | WC | TCP |
| $F_8$ | * | 000* | WC | Both | WC |

After port and protocol fields are classified based on their values, the comparison results of a field can have the following three relationships:

1. Disjoint: When the two compared port field values belong to HI and LO, respectively, we can determine that the comparison results of the two filters in this field must be an empty set because the values in these two classes are disjoint. For example, we cannot find a value that is located in both regions [1024:65,535] and [0:1023]. With the same method, except the case in which the protocol value is a wildcard, different protocols are disjoint.

2.  Overlap: For port and protocol fields, when one of the two compared fields is a wildcard or the values of both two compared protocol fields are TCP or UDP, we can determine that the comparison results of this field must not be an empty set because the wildcard indicates that the value covers all ranges. If it is compared with any other classes, they will certainly overlap.

3.  Requiring exact comparison: In this relationship, we must fetch the complete information of this field for an exact comparison to determine the final comparison results. For example, if the two compared port values are exact values 23 and 80, respectively, which are classified as LO, we know that they are disjoint only after an exact comparison. With the same method, if the two compared port values are an exact value 80 and a range [0:1023], respectively, we know that they are overlapping only after exact comparison. Similarly, if the protocol field belongs to Other, further exact comparison is required to determine the result.

Table 7 lists all comparison combinations of two filters *F* and *G* in a single non-prefix field. When a single non-prefix field comparison is extended to all non-prefix fields comparison, we can list the classes of all comparison combinations of non-prefix fields based on the three conflicting combinations described in Section 4.1 because each field has several types of comparison results. We can then analyze the comparison results to identify those combinations that are known without exact comparison.

**Table 7.** All comparison combinations of two filters *F* and *G* in a single non-prefix field.

| Relationship | Disjoint | | Overlap | | Requiring Exact Comparison | |
|---|---|---|---|---|---|---|
| | **Port Field** | **Protocol Field** | **Port Field** | **Protocol Field** | **Port Field** | **Protocol Field** |
| **(F, G)** | (HI, LO) (LO, HI) | (TCP, UDP) (TCP, Other) (UDP, TCP) (UDP, Other) (Other, TCP) (Other, UDP) | (WC, *) (*, WC) | (WC, *) (*, WC) (TCP, TCP) (UDP, UDP) | (HI, HI) (HI, Both) (LO, LO) (LO, Both) (Both, HI) (Both, LO) (Both, Both) | (Other, Other) |

### 4.3. Relationships of Non-Prefix Field Combined Comparison

After defining the comparison results of two filters in a single non-prefix field, all comparison result combinations of all non-prefix fields can be classified into the following three sets:

1.  Disjoint: In these compared non-prefix fields, the comparison results of at least one field are disjoint. When the compared non-prefix fields belong to the combination of this type, no conflict exists after combined detection regardless of the combination to which the comparison results of the prefix fields of filters *F* and *G* belong. For example, the non-prefix field combinations of filter pair $(F_1, F_6)$ are (LO, WC, TCP) and (HI, WC, UDP). Filter pair $(F_1, F_6)$ is not in conflict because the comparison results in fields Source port and Protocol are disjoint.

2.  Overlap: The comparison results of these compared non-prefix fields all overlap. When these compared non-prefix fields belong to the combinations of a set that is the same as the 3*Dset*1 set listed in Table 3 (i.e., the comparison result of prefix fields of filters *F* and *G* is Combination 1 defined in Section 4.1,), it will conflict with the non-prefix field combination in this set after combined detection. In Section 4.1, we discussed the relationship between these three sets: 3*Dset*1, 3*Dset*2, and 3*Dset*3. Therefore, in this set we can also find a set of three field combinations that matches 3*Dset*2 and 3*Dset*3. The set of non-prefix field combinations that these two sets match is derived as follows. 3*Dset*2: In the non-prefix fields of *F* and *G* comparison, at least one field *s* exists and the value of *F*[*s*] is a wildcard, but not the value of *G*[*s*] (e.g., the destination port field of filter pair $(F_2, F_4)$ are WC, LO in Table 6). 3*Dset*3: In

the non-prefix fields of *F* and *G* comparison, at least two fields *s* and *t* exist, the value of *F*[*s*] is wildcard, and the value of *G*[*s*] is not; or the value of *F*[*t*] is not wildcard and the value of *G*[*t*] is wildcard (e.g., the destination port and protocol fields of filter pair ($F_7$, $F_8$) are (WC, Both) and (TCP, WC) in Table 6).

3. Requiring exact comparison: In these compared non-prefix fields, the comparison results of at least one field require exact comparison, whereas the comparison results of other fields overlap. When the compared non-prefix fields belong to the combination of this set, exact comparison is required after combined detection regardless of the combination to which the comparison results of the prefix fields of filters *F* and *G* belong. For example, if the combinations of filter pair ($F_1$, $F_5$) are (LO, WC, TCP) and (LO, WC, WC), the comparison results of filter pair ($F_1$, $F_5$) can be determined only after exact comparison of the source port field.

To distinguish among three sets 3*Dset*1, 3*Dset*2, and 3*Dset*3, we divide the overlapped set of three compared fields into three disjoint sets. The method selects the combinations that satisfy the conditions of 3*Dset*3 set as distinguished from 3*Dset*1 set and includes them in an independent set 3*Dset*3'. It then includes the combinations that meet the conditions of the 3*Dset*2 set into an independent set 3*Dset*2'. The remaining combinations in the 3*Dset*1 are treated as an independent set 3*Dset*1'. The combinations of the three compared fields can be classified into five parts, as shown in Figure 2.

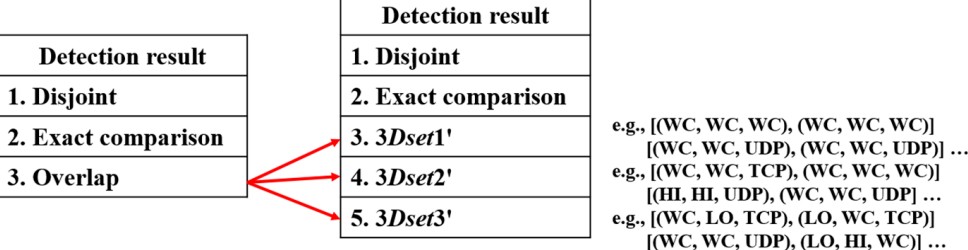

**Figure 2.** Five results of combined comparison.

From the aforementioned five combined comparison results and the examples provided, we can predict the situation of each non-prefix field comparison based on classification of the port and protocol values. When the comparison result is disjoint or overlaps, we can determine whether the two compared filters conflict without requiring exact comparison. Based on our designed pre-computed information, the filter pairs that do not require exact comparison can be filtered out to reduce the frequencies of memory access caused by reading complete information and by exact comparison. As a consequence, the overall conflict detection can be performed faster.

*4.4. Lookup Table Construction*

Based on the analysis in Section 4.3, we can divide all comparison results of non-prefix fields into five cases. In order to implement our proposed algorithm, for each conflict case, we create a comparison result lookup table by listing each matching result. To minimize the memory requirement for the comparison result lookup table, we use binary code to represent the list of classes for port and protocol fields, and list corresponding values for the results, as shown in Tables 8 and 9, respectively. The binary codes of two filters are combined in serial to form a 12-bit-length index value. For example, the index value is (100001, 000001) after filters $F_1$ and $F_2$ in Table 6 are combined. Each index value indicates the information content of the non-prefix fields of the two filters. According to the conflicting comparison combinations defined in Table 3, the corresponding values of the comparison results can be defined. During non-prefix field conflict detection, we determine whether an exact comparison action is required based on the comparison results, or determine whether conflicts exist based on the comparison results of prefix fields. Table 10 provides an example of a comparison result lookup table. Finally, we use Figure 3

to show the overall process of our 5D conflict detection flow. When two compared filters *F* and *G* are in conflict during prefix field comparison, we must read and combine the binary code serials of *F* and *G*, perform an index value search to obtain the corresponding value for a comparison result, and finally combine the detection process to obtain the final results. If the final results are disjoint, then exact comparison is not required in order to shorten the overall conflict detection time. For example, during conflict detection of filters $F_2$ and $F_3$ in Table 3, the comparison result of prefix fields is equal, then prefetching the binary code serials of filters $F_2$ and $F_3$ is (000001, 001000) which represent as (WC, WC, TCP) and (WC, LO, WC). The comparison result of index value (000001, 001000) is 100 which belongs to 3*Dset*3'. Finally, we determine that filters $F_2$ and $F_3$ are in conflict without an exact comparison because the comparison result satisfies Combination 3 defined in Section 4.1. Similarly, we determine that filters $F_6$ and $F_7$ in Table 3 do not have conflict without an exact comparison because the comparison using (010010, 000001) as the index value of the lookup table result is 000. Here, we conclude that the greater the number of exact comparisons that are filtered, the faster conflict detection will be.

**Table 8.** Code of port and protocol field.

| Port Field | Code | Protocol Field | Code |
|:---:|:---:|:---:|:---:|
| WC | 00 | WC | 00 |
| HI | 01 | TCP | 01 |
| LO | 10 | UDP | 10 |
| Both | 11 | Other | 11 |

**Table 9.** Code of the combined comparison results.

| Comparison Result | Code |
|:---:|:---:|
| Disjoint | 000 |
| Requiring exact comparison | 001 |
| Set 3*Dset*1' | 010 |
| Set 3*Dset*2' | 011 |
| Set 3*Dset*3' | 100 |

**Table 10.** The comparison result lookup table.

| (Filter *F*, Filter *G*) | Code of Comparison Result |
|:---:|:---:|
| (000000, 000000) | 011 |
| (000000, 000001) | 011 |
| (000000, 000010) | 011 |
| ⋮ | ⋮ |
| (000001, 001000) | 100 |
| ⋮ | ⋮ |
| (010010, 000001) | 000 |
| ⋮ | ⋮ |
| (111111, 111101) | 000 |
| (111111, 111110) | 000 |
| (111111, 111111) | 001 |

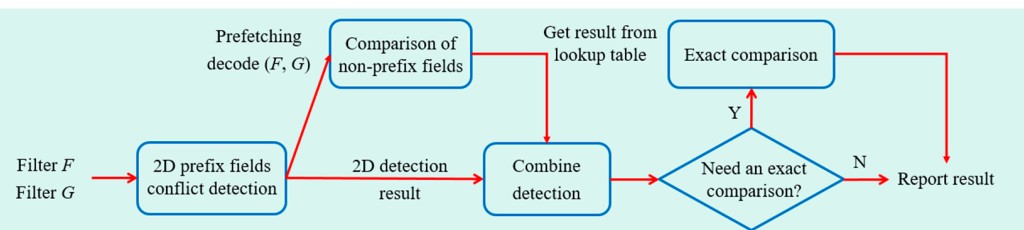

**Figure 3.** The process flow of 5D conflict detection.

The size of the entire lookup table is $2^{12}$ (the number of index values) $\times$ 3 (the number of bits required by the matching result) = $3 \times 2^{12}$ bits. In other words, only 1.5 KB is needed. For each filter, only an additional 6 bits of memory space are required to store its code information.

## 5. Experimental Results

In this section, we evaluate the performance of our proposed algorithm for 5D conflict detection. In the experiment, the tested filter databases were synthesized by Class-Bench [33], which is widely used in packet classification and conflict detection research for test simulation. ClassBench can generate filter databases with different properties by using 12 seed files, including three practical application types: access control lists (ACL), firewalls (FW), and IP chains (IPC). For each seed file, we generate three filter sets containing 5 K, 10 K, and 20 K filters, respectively. The performance metrics is the average conflict detection time in microseconds (µs) required for each filter. In our simulation experiment, we compared our proposed algorithm with those existing 5D conflict detection of two other types. The first part of the experiment tested the difference in performance between the non-prefix field comparison using our algorithm and the non-prefix field comparison using exact comparison, under the condition that the existing 2D conflict detection algorithm was extended to a 5D conflict detection algorithm without any pre-computed information. Because the comparison focused on the efficiency of detecting non-prefix fields, no noticeable difference was observed between our algorithm and any existing 2D conflict detection algorithm for comparison of prefix fields. We then compared our algorithm with the *FastDetect* algorithm [6] because the authors have shown how their algorithm performs conflict detection for non-prefix fields without additional data structure, and it is helpful to compare the performance between the existing 2D algorithms and our algorithm for non-prefix fields. The second part of the experiment compared our algorithm with the existing detection algorithm which can be extended from 2D detection to 5D detection. To support conflict detection for non-prefix fields, this latter type of algorithm requires a new auxiliary data structure. Therefore, we tested the differences in time and space performance between non-prefix field conflict detection using the original comparison mode and that using our algorithm. Here, the SBV algorithm was used for comparison [7]. All algorithms were implemented in C++ and benchmarked on an Intel Core i5–4440 3.1-GHz processor with 12 GB memory.

### 5.1. Filtering Ratio of the Comparison Result Lookup

We first examine how much the lookup table mentioned above can help in reducing exact comparisons. The results listed in Table 11 show that considerable differences exist between the average numbers of performing non-prefix field comparisons for every filter in a range of 20 K databases. The trend revealed was: the fewer conflicting pairs in the database, the lower the average exact comparison for filters. On average, exact comparison was performed for filters in the ACL5 database less than once, whereas it was performed almost 2500 times for filters in the FW5 database. The numbers required to perform exact comparison for filters in the ACL databases were far fewer than those for the FW databases, while the difference between the two IPC databases was also considerable.

The difference of the comparison result distribution between different databases was also considerable. The three rightmost columns show the percentage distributions of the results of non-prefix field comparisons for all databases. As defined in Section 4.3, the sum of the percentages of set Disjoint and set Conflict indicates the probability of knowing detection results after a single comparison conducted during filter conflict detection. By contrast, the percentage of the set requiring exact comparison indicates the probability of detection results that are uncertain in this stage and require exact comparison. For example, our algorithm reduces the average number of exact comparisons performed for each filter in the ACL2 database from 250.98 to 27.43 (250.98 $\times$ 10.93% = 27.43), which proves that our

algorithm features high filtering efficiency. The percentage of confirmed conflict includes the three combinations that will conflict after combined detection is performed.

**Table 11.** Performance evaluation for 20 K filter databases.

| Databases | Statistics | | Average Number of Non-Prefix Field Comparisons per Filter | Percentage Distribution of the Results of Non-Prefix Field Comparisons | | |
|---|---|---|---|---|---|---|
| | Number of Filters | Number of total Non-Prefix Field Comparisons | | Disjoint | Conflict | Requiring Exact Comparison |
| ACL1 | 19,912 | 188,707 | 9.48 | 23.38% | 46.46% | 30.16% |
| ACL2 | 18,674 | 4,686,814 | 250.98 | 19.45% | 69.62% | 10.93% |
| ACL3 | 19,049 | 3,576,805 | 187.77 | 64.70% | 12.04% | 23.25% |
| ACL4 | 19,221 | 3,022,216 | 157.24 | 65.46% | 11.06% | 23.48% |
| ACL5 | 13,585 | 994 | 0.07 | 75.96% | 0.00% | 24.04% |
| FW1 | 18,643 | 38,711,274 | 2076.45 | 67.13% | 20.45% | 12.42% |
| FW2 | 19,254 | 24,533,881 | 1274.22 | 0.00% | 68.63% | 31.37% |
| FW3 | 17,743 | 39,051,434 | 2200.95 | 69.93% | 13.10% | 16.97% |
| FW4 | 17,333 | 23,916,573 | 1379.83 | 59.04% | 19.57% | 21.39% |
| FW5 | 17,470 | 42,750,196 | 2447.06 | 60.57% | 21.78% | 17.65% |
| IPC1 | 19,477 | 3,532,940 | 181.39 | 32.49% | 50.13% | 17.38% |
| IPC2 | 20,000 | 12,162,906 | 608.15 | 0.00% | 66.90% | 33.10% |

*5.2. Comparing Time Performance with Original 2D Detection Algorithms*

In this subsection, we test the time performance difference between our algorithm and the *FastDetect* algorithm using exact comparison in non-prefix field conflict detection. Table 12 shows the average conflict detection time required for each filter in different–sized databases. Experimental data shows that our algorithm was no less than 10% faster than the general exact comparison methods in non-prefix field conflict detection for databases of different types and sizes. This also proves that filtering out the time in which exact comparison is not required will improve the time performance. Because the average time of performing exact comparison for every filter in databases ACL1 and ACL5 was minimal, the effect of the filtering algorithm was not obvious. The required time was almost the same for the filter conflict detection of these two databases. The performance of the ACL2 database was better than the other ACL databases except for ACL1 and ACL5 due to ACL2 containing the lowest ratio of requiring exact comparison sets. Similarly, FW1 performed better than most FW databases. For IPC databases, the huge difference in the number of exact comparisons required between databases IPC1 and IPC2 also showed that our algorithm performs better.

Our method requires only small memory space to extend the original 2D conflict detection to 5D conflict detection for specific performance improvement. Since an additional data structure is not required, it prevents the possibility that the existing data structure will change with frequent addition of dynamic filters and application updates, and reduces the impact to the overall performance.

*5.3. Comparing Time Performance with Extended 5D Algorithms*

In this subsection, we discuss the difference in performance between our proposed algorithm and the comparison method that the SBV algorithm employs for non-prefix field conflict detection. Because the original SBV algorithm cannot detect the conflicts defined in [6], we refer to the method provided in [10] to adjust the SBV algorithm so that it can detect these conflicts. Table 13 shows the average conflict detection time for every filter in different-sized databases. The data shows that our algorithm is 10% faster than the SBV algorithm in most databases. For databases FW1, FW3, and FW5, our algorithm does not improve the efficiency as obviously as do the other databases because the characteristics of the SBV algorithm include the filtering function. The SBV algorithm creates a binary search

trie for every to-be-compared field and the algorithm searches the target node of the binary search trie during detection. When *k* compared fields exist, the AND logic comparison results of *k*-bit vectors collected finally may overlap the detecting filters. In other words, the final results may exclude the sets that do not overlap the detecting filters. The effect is the same as the disjoint results in our comparison result lookup table. Therefore, if the ratio of disjoint sets is high in the database, the SBV algorithm can also filter out a considerable number of filter sets that do not require exact comparison. Table 11 shows the high ratios (more than 60%) of disjoint sets in databases FW1, FW3, and FW5. Similarly, the performance of databases ACL3 and ACL4 is less than that of other ACL databases. Although database ACL5 contains a high ratio of disjoint sets, the impact on its performance is not obvious because this database requires only few instances of exact comparison, on average. By contrast, if the ratio of confirmed conflicts in the comparison result query is high and the ratio of disjoint sets is relatively low (less than 40%), the SBV algorithm cannot filter out most filter sets that do not require exact comparison. Therefore, our algorithm performs much better than the SBV algorithm with the databases ACL1, ACL2, FW2, IPC1, and IPC2.

**Table 12.** Average detection time required to detect conflicts for databases of three sizes.

| Databases | 5K | | | 10K | | | 20K | | |
|---|---|---|---|---|---|---|---|---|---|
| | Average Detection Time | | Speedup | Average Detection Time | | Speedup | Average Detection Time | | Speedup |
| | Fast | Ours | | Fast | Ours | | Fast | Ours | |
| ACL1 | 0.45 | 0.36 | 20.00% | 0.69 | 0.62 | 10.14% | 0.97 | 0.75 | 22.68% |
| ACL2 | 8.15 | 6.83 | 16.20% | 15.53 | 12.48 | 19.64% | 28.59 | 23.24 | 18.71% |
| ACL3 | 7.77 | 6.49 | 16.47% | 12.55 | 10.7 | 14.74% | 19.06 | 16.49 | 13.48% |
| ACL4 | 5.77 | 5.11 | 11.44% | 8.49 | 7.55 | 11.07% | 16.21 | 14.44 | 10.92% |
| ACL5 | 0.11 | 0.09 | 18.18% | 0.13 | 0.09 | 30.77% | 0.14 | 0.1 | 28.57% |
| FW1 | 88.28 | 67.55 | 23.48% | 362.74 | 232.61 | 35.87% | 481.74 | 400.73 | 16.82% |
| FW2 | 45.29 | 40.33 | 10.95% | 91.31 | 79.31 | 13.14% | 210.65 | 189.07 | 10.24% |
| FW3 | 191.88 | 131.88 | 31.27% | 441.27 | 353.3 | 19.94% | 691.06 | 609.51 | 11.80% |
| FW4 | 50.53 | 42.75 | 15.40% | 140.17 | 119.9 | 14.46% | 332.15 | 286.39 | 13.78% |
| FW5 | 159.38 | 136.22 | 14.53% | 464.03 | 395.66 | 14.73% | 879.9 | 746.2 | 15.19% |
| IPC1 | 4.7 | 4.17 | 11.28% | 9.93 | 8.8 | 11.38% | 19.69 | 17.27 | 12.29% |
| IPC2 | 45.79 | 35.58 | 22.30% | 109.85 | 93.73 | 14.67% | 302.72 | 234.91 | 22.40% |

**Table 13.** Average detection time required to detect conflicts for databases of three sizes.

| Databases | 5K | | | 10K | | | 20K | | |
|---|---|---|---|---|---|---|---|---|---|
| | Average Detection Time | | Speedup | Average Detection Time | | Speedup | Average Detection Time | | Speedup |
| | SBV | Ours | | SBV | Ours | | SBV | Ours | |
| ACL1 | 6.81 | 5.26 | 22.76% | 13.59 | 11.06 | 18.62% | 26.94 | 22.13 | 17.85% |
| ACL2 | 7.82 | 5.72 | 26.85% | 15.11 | 11.4 | 24.55% | 27.61 | 22.11 | 19.92% |
| ACL3 | 7.46 | 5.95 | 20.24% | 13.89 | 11.5 | 17.21% | 26.86 | 22.87 | 14.85% |
| ACL4 | 7.43 | 6.01 | 19.11% | 14.02 | 11.41 | 18.62% | 26.86 | 22.65 | 15.67% |
| ACL5 | 4.55 | 3.39 | 25.49% | 9.9 | 7.83 | 20.91% | 18.21 | 15.27 | 16.14% |
| FW1 | 8.45 | 7.27 | 13.96% | 15.82 | 14.28 | 9.73% | 31.7 | 28.27 | 10.82% |
| FW2 | 8.83 | 6.33 | 28.31% | 15.93 | 12.53 | 21.34% | 30.06 | 25.24 | 16.03% |
| FW3 | 7.89 | 7.14 | 9.51% | 15.76 | 14.13 | 10.34% | 30.99 | 27.82 | 10.23% |
| FW4 | 8.97 | 7.17 | 20.07% | 15.82 | 13.3 | 15.93% | 28.96 | 25.23 | 12.88% |
| FW5 | 8.83 | 7.51 | 14.95% | 15.99 | 14.67 | 8.26% | 33.65 | 28.32 | 15.84% |
| IPC1 | 7.59 | 5.85 | 22.92% | 13.86 | 11.43 | 17.53% | 26.41 | 22.9 | 13.29% |
| IPC2 | 8.82 | 6.19 | 29.82% | 15.87 | 12.51 | 21.17% | 28.03 | 24.44 | 12.81% |

Remarkably, to perform 5D conflict detection, the SBV algorithm must additionally construct binary search tries and set bit vectors for these non-prefix fields. Besides additional memory space, constructing a corresponding data structure also increases the execution time. We list the execution time and memory space that the SBV algorithm requires in order to construct the additional data structure in the 20K filter databases. Table 14 shows that constructing an additional data structure for non-prefix fields costs considerable time and memory compared to our proposed algorithm, which cost little time and memory to set up 6-bit information for each filter and 1.5 KB lookup table. For applications that require frequent filter updates, the performance impact of the SBV algorithm is even greater. For example, each filter costs an average of 30.79 microseconds to build auxiliary information. In other words, it takes 30.79 microseconds to update each filter which is more than the time it takes to detect all conflicts in FW4. All the algorithms that use the trie-based data structure are affected in the same way. That is the reason we only compared our algorithm with the SBV algorithm.

**Table 14.** Cost of the SBV algorithm to build auxiliary information with 20K filter databases.

| Databases | Number of Filters | Total Time (μs) | Average Build Time | Space (KB) |
|---|---|---|---|---|
| ACL1 | 19,912 | 104,103.40 | 5.23 | 3197.71 |
| ACL2 | 18,674 | 110,627.00 | 5.92 | 680.75 |
| ACL3 | 19,049 | 172,434.20 | 9.05 | 4625.60 |
| ACL4 | 19,221 | 166,306.80 | 8.65 | 5388.98 |
| ACL5 | 13,585 | 80,651.80 | 5.94 | 752.05 |
| FW1 | 18,643 | 190,337.60 | 10.21 | 1386.76 |
| FW2 | 19,254 | 180,186.80 | 9.36 | 214.29 |
| FW3 | 17,743 | 157,403.60 | 8.87 | 1183.20 |
| FW4 | 17,333 | 533,745.20 | 30.79 | 1711.34 |
| FW5 | 17,470 | 129,372.20 | 7.41 | 1141.49 |
| IPC1 | 19,477 | 117,780.00 | 6.05 | 2037.23 |
| IPC2 | 20,000 | 53,946.80 | 2.70 | 146.86 |

## 6. Conclusions

Conflict detection ensures the correctness of packet classification and has received considerable attention in recent years. However, most conflict detection algorithms detect conflicts for only prefix fields. For greater practicality, the non-prefix fields should be considered. In this study, our proposed algorithm focuses on the conflict detection of non-prefix fields and is used either to extend an existing 2D conflict detection algorithm to support fast 5D conflict detection, or to increase the throughput of an existing 5D conflict detection algorithm. As the experimental data revealed, our proposed algorithm showed significantly better performance than the *FastDetect* and SBV algorithms when detecting conflicts for filter databases of different sizes and types. Since the SBV algorithm can also filter out in advance those filter sets that do not conflict, the performance of our algorithm is clearly not better than the SBV algorithm when detecting some FW-type filter databases. However, the SBV algorithm requires additional running time and memory cost for additional auxiliary data structure; when the number of fields in the filter comparison increases or the update operations are frequent, these additional costs will affect the performance of the SBV algorithm. With our algorithm, the additional cost can be avoided because the memory demand is minimized.

**Author Contributions:** Conceptualization, C.-L.L. and Y.-C.C.; Data curation, G.-Y.L.; Formal analysis, G.-Y.L.; Investigation, G.-Y.L.; Methodology, G.-Y.L.; Project administration, C.-L.L. and Y.-C.C.; Resources, Y.-C.C.; Software, G.-Y.L.; Supervision, C.-L.L. and Y.-C.C.; Validation, G.-Y.L.; Writing—original draft preparation, G.-Y.L.; Writing—review and editing, C.-L.L. and Y.-C.C. All authors have read and agreed to the published version of the manuscript.

**Funding:** This research was funded by the Ministry of Science and Technology of Taiwan (MOST 104-2221-E-182-005 and 110-2221-E-182-010) and Chang Gung Memorial Hospital (BMRP 942).

**Data Availability Statement:** Not applicable.

**Conflicts of Interest:** The authors declare no conflict of interest.

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
