# Peer review of "Fast Conflict Detection for Multi-Dimensional Packet Filters"

_algorithms, doi:10.3390/a15080285_

Round 1

Reviewer 1 Report

The paper presents a “mechanism” (see below the notes about terminology confusion) for network packet filtering based on 5 fields. Overall, the paper is well-written except some minor spelling mistakes. For example, “fields of P matches all fields of F”, “each filed is 4 bits”, and so on.

The Reviewer has several concerns about the paper which, hopefully, can be quickly fixed.

First, the Authors should clearly define the contribution of the paper. Is it an approach, method, algorithm, or something else? For example, in the Abstract the Authors write “This study analyzes…” then “Finally, a lookup table is created …”. Later the Authors write “This study proposes an efficient detection mechanism for non-prefix fields of filters.” Hence, the terms and definitions should be sorted out in the paper.

Second, the Authors should clearly explain why “… most algorithms cannot be extended for 5D conflict detection.” and why they select FastDetect and SBV for comparison. Are there any other relevant candidates that can be used for comparison? If no, this should be clearly explained. Otherwise, a relevant algorithm must be included for comparison (currently we have only 2 without a convincing justification).

Finally, in the Related Work section, the Authors provide asymptotic time/space estimations for the algorithms. The paper could be stronger, if the Authors can provide similar estimates for their “mechanism” / ”approach” / … (please, decide) or part of their contribution that is amenable for such asymptotic estimates.

Reviewer 2 Report

Thank you for providing me with the opportunity to read “Fast Conflict Detection for Multi-Dimensional Packet Filters”. I have the following comments:

·         Consider adding 1-2 more keywords to the keyword section.

·         The paper needs serious support from the literature. The lack of a state-of-the-art review is reflected in the references of the paper, which are very low. Line 27-42 needs references for all strong statements made in the associated paragraphs.

·         The paper has a lot of abbreviations; please add a table for abbreviations.

·         Please highlight the objectives of the study in a numbered format. Currently, these are not clear and embedded in the paper.

·         In the novelty portion of the paper, please add support from the literature. This should focus on the key issues that you are addressing through the current study.

·         The references are wrongly numbered, or some are missing in between. For example, line 130 ref 6 appears after ref 1; where are other references? Line 130-143 need support from the literature.

·         The related work section should be divided into sub-sections to better present the work and ease the readability of the paper.

·         A detailed method section is needed in the paper. Rename section 3 as a method and add a compressible overview of the paper here. A figure should be added here to graphically explain the structure of the paper and the techniques used along with the data collection mechanisms. The same should be elaborated on in the following sections.

·         Why is 5D conflict method used in the paper? Justification is needed for this and other techniques used in the paper. This should be explained in the new method section.

·         Similarly, more information is needed about the datasets used in the paper.

·         Table 6 needs more explanation. What is the key takeaway here? The author needs to explain the key message here. Alternatively, remove the Table if it is not adding any value; please consider deleting it.

·         Similarly, elaborate more on the key messages of Tables 10 and 11.

·         The paper needs a detailed discussion section where the authors need to focus on the key takeaways and how the current study compares with existing studies. What results are given for the same problem by other published works, and then discuss how your study is comparable to or better than these. A table can be added to compare the results.

·         In the conclusion, present the key results in 2-3 lines to highlight what is important in the study.

·         What are the implications of this study? Please add to the conclusion section.

Round 2

Reviewer 2 Report

Thank you for addressing my comments.